# Research on UAV Swarm Network Modeling and Resilience Assessment Methods

**DOI:** 10.3390/s24010011

**Published:** 2023-12-19

**Authors:** Xinjue Zhang, Jixin Liu

**Affiliations:** 1College of Civil Aviation, Nanjing University of Aeronautics and Astronautics, Nanjing 211106, China; zxj1273808869@163.com; 2National Key Laboratory of Air Traffic Flow Management, Nanjing 210016, China

**Keywords:** UAV swarm, air traffic management, complex network, independent weight factor method, resilience assessment

## Abstract

The traditional UAV swarm assessment indicator lacks the whole process description of the performance change after the system is attacked. To meet the realistic demand of increasing resilience requirements for UAV swarm systems, in this paper, we study the modeling and resilience assessment methods of UAV swarm self-organized networks. First, based on complex network theory, a double layer coupled UAV swarm network model considering the communication layer and the structure layer is constructed. Then, three network topological indicators, namely, the average node degree, the average clustering factor, and the average network efficiency, are used to characterize the UAV swarm resilience indicators. Finally, the UAV swarm resilience assessment method, considering dynamic evolution, is designed to realize the resilience assessment of the UAV swarm under different strategies in multiple scenarios. The simulation experiments show that the UAV swarm resilience assessment, considering dynamic reconfiguration, has a strong correlation with the network structure design.

## 1. Introduction

### 1.1. Background of the Study

With the continuous advancement of Chinese low-altitude airspace management reform, the demand for civil UAV flights has shown a trend of rapid growth and has been widely used in commercial, public, military, tourism, and sports fields [1]. The construction of the 5G network has led to the rapid development of the communication field, and the communication capability between UAVs has been significantly improved, so that UAV swarms in the future will have a wider range of application prospects than a single UAV [2]. Compared with a single UAV, UAV swarms have higher group efficiency and better survivability, while group cooperative operations also have higher mobility. Therefore, UAV swarms have wider applications in joint reconnaissance, search and rescue, and cooperative operations [3].

In recent years, many scholars have carried out systematic research on UAV swarms and other multi-intelligence systems [4,5], and the research direction mainly focuses on UAV swarm topology design [6,7,8], task assignment [9,10,11], path planning [12,13,14,15,16], cooperative control [17,18,19], and planning scheduling [20,21]. However, most of these studies assume that the UAV swarms are in a disturbance-free situation and do not consider the performance of UAV swarms in abnormal flight conditions.

### 1.2. Related Works

In their research about the analysis and modeling of complex systems such as UAV swarms, scholars have proposed theoretical methods such as agent-based modeling, the Petri net method, system dynamics, and complex networks. These methods are all very effective in modeling complex systems and can be adapted to the scenarios and applications of different systems, respectively. Among them, one of the main advantages of complex network theory is that it can identify the universal properties in the link structure of the basic units of the system [22]. Therefore, complex networks are often used to represent all kinds of complex group system structures, and many scholars have mapped the operational situation of UAV swarms to complex network models and analyzed the damage to the network structure of UAV swarms through complex network indicators. In terms of UAV swarm network modeling, Wu [23] mapped the number of UAVs and the number of their neighboring UAVs to the concepts of several nodes and the degree of nodes in a complex network, which is used as a measure of the transient capability of UAV swarms in carrying out joint reconnaissance missions. In practical applications, even if some individuals in the UAV swarm are damaged, the whole UAV swarm must ensure the successful execution of the mission, thus putting higher requirements on the resilience of the UAV swarm network.

In recent research, several scholars have proposed resilience measurement methods applicable to UAV swarms. Tran [24] introduced a generic resilience metric for assessing information exchange networks, utilizing the BA model (Barabási–Albert model) to construct a complex network that grows by adding nodes. Building upon Tran’s work, Cheng C. [25] presented an enhanced resilience evaluation metric, formulating a total resilience measure as the sum of absorptive capacity and recovery ability. Weight coefficients were assigned to enhance formula flexibility based on various system requirements. This research application was demonstrated in the context of joint reconnaissance missions for UAV swarms. Bai G. [26] contended that Tran’s resilience coefficient, calculated as the ratio of the corresponding capabilities before and after disruption, may not be suitable for UAV swarms with mission attributes. Considering the goal of UAV swarms is to accomplish assigned tasks, the study proposed a new coefficient by modifying the critical performance factors function for UAV swarms. Zhang P et al [27]. defined UAV nodes in three different states: normal, overloaded, and failed. The ability of UAV swarms to execute tasks was defined based on the ratio of working nodes to failed nodes. Subsequently, an indicator measuring the resilience of UAV swarms was derived. Various methods for network modeling and resilience measurements are shown in Table 1.

### 1.3. Contributions

However, the current research still has the following deficiencies, which are mainly reflected in the following: first, in the current research on self-organized network models of UAV swarms based on complex networks, some of the simple network models do not apply to the actual environment due to the over-idealized requirements on the links between the UAVs; second, most of the currently proposed resilience indicators are improvements to an existing resilience assessment methodology, lacking a more objective quantitative definition of resilience, and lacking a horizontal all-round objective consideration; third, most of the current research focuses on the recovery strategy of UAV swarms after perturbation and seldom involves the analysis of the performance change of UAV swarms during the process of disturbance.

This paper focuses on the dynamic evolution of UAV swarms post-damage events. It constructs a self-organized network model, considering the double layer coupling of the communication and structural layers. The study establishes a comprehensive resilience assessment metric method using various network topology indices. Additionally, it designs a resilience assessment method for UAV swarms that considers dynamic evolution. This approach enables resilience assessment in multiple scenarios and under different strategies. The main contributions and innovations of this study are as follows:
We take into consideration the distinct characteristics of the self-organizing network communication layer and structural layer within UAV swarms, along with the dynamic reconstruction and information correlation properties. It proposes a double layer coupled model for UAV swarms based on complex networks;The comprehensive resilience measurement for UAV swarms introduced in this study incorporates three network topology indicators. These indicators, when combined, offer a thorough and objective assessment of the overall network structure, local clustering, and information transmission efficiency. This approach achieves multidimensional extraction of comprehensive metrics;This study introduces two distinct resilience assessment methods. The resilience assessment, considering the dynamic reconstruction of UAV swarms, analyzes differences in resilience among three network structures under different strategy combinations. The resilience assessment, considering information correlation, provides resilience curves for three network structures and analyzes the impact of different recovery rates on network resilience.


## 2. Modeling of Self-Organizing Networks of UAV Swarm

In this section, based on the modeling method of complex network theory, the dynamic evolution characteristics of UAV swarms are analyzed, and a double layer coupled UAV swarm self-organizing network model considering the communication layer and the structural layer is constructed.

### 2.1. Complex Network Based Double Layer Coupled UAV Swarm Modeling

The method of modeling UAV swarms based on complex networks is to consider individual UAVs in the cluster as nodes and the connections between the UAVs as edges, and the dynamic changes of UAV swarms can be regarded as the changes of nodes and connected edges in the complex network model [28]. The dynamic evolution of the UAV swarm is modeled by removing nodes or edges randomly or directionally based on the characteristics of nodes such as degree, median, and agglomeration factor [29] at the stage when the swarm suffers from disturbances or damages.

In this part, the UAV swarm system is constructed as a double layer coupled network model, considering the structural and communication layers. In the structure layer, each UAV itself is considered a node, and the physical distance between UAVs that satisfy communication distances is regarded as an edge; in the communication layer, each UAV is regarded as a communication point, and the communication links between UAVs can be regarded as edges. When constructing the network model, the neighbor matrix ${A=[a}_{ij}{]}_{M\times M}$ is used to represent the link relationship of individual nodes. If the node *i* and the node *j* have connected edges, the matrix $a_{ij}=a_{ji}=1$, otherwise $a_{ij}=a_{ji}=0$.

A simplified schematic diagram of the initial UAV swarm self-organizing network structure is established based on the complex network *G* = {*V*, *E*}, where *V* denotes nodes and *E* denotes connected edges. The UAVs are abstracted as structural nodes and communication nodes, respectively, and there is an association between the two nodes that each UAV is transformed into, and this is generalized to the flight network composed of 10 UAVs to construct a double layer UAV swarm network between the structural layer and the communication layer, as shown in Figure 1.

### 2.2. UAV Evolution Analysis and Modeling

When external threat events perturb the UAV swarm network system, it undergoes a dynamic evolution process. This process includes the dynamic reconfiguration of nodes and the association of information. If the cascade failure phenomenon occurs, where large-scale nodes fail successively, it results in the communication interruption of the entire network. This has a significant impact on the stability of the system. Therefore, the prerequisite for studying the resilience of the UAV swarm system is to analyze the dynamic evolution process and conduct modeling.

#### 2.2.1. Dynamic Reconfiguration Analysis and Modeling

The dynamic reconfiguration process is the process by which a system quickly recovers its capabilities and rebuilds its structure and performance in the event of an attack. In complex network theory, dynamic reconfiguration denotes the situation in which a network system re-links edges after the phenomenon of node loss. For the UAV swarm network, when the UAV swarm suffers damage that leads to node failure, there may exist some isolated nodes or a cluster of island nodes with no edges between them and other nodes that consist of a small number of components of the internal network. To restore communication, the failed nodes must link to other UAV nodes. Randomly link to neighboring nodes or selectively link to other remaining nodes in the network based on the node’s characteristic metrics, such as degree, median, and agglomeration factor.

In the UAV swarm damage phase, when a UAV is attacked and its communication fails, the failed node and the edges connected to it should be removed from the network in the communication layer, as shown in Figure 2a; in the UAV swarm recovery phase, the red edges are the edges for the re-linking of the remaining nodes, as shown in Figure 2b. When a UAV is physically damaged and fails, the structural and communication nodes of the UAV in the network will fail simultaneously, as shown in Figure 2c; Figure 2d represents the network recovery at the structural and communication layers. The processes (b) and (d) in Figure 2 are the dynamic reconfiguration of the UAV swarm.

#### 2.2.2. Information Correlation Analysis and Modeling

In the communication layer of a UAV swarm, nodes have information correlated with each other, i.e., the failure of a communication node directly affects the remaining nodes connected to it. Since the wireless communication layer network of UAV swarms is the basis for receiving commands from ground sites and accomplishing established tasks, the communication layer of UAV swarms usually becomes a key attack layer for viruses and malicious programs. According to the propagation characteristics of viruses, the attacked node may cause the failure of the remaining nodes through the coupling relationship between the nodes and then produce a cascade effect to cause the collapse of most of the nodes in the UAV swarm or the whole network, i.e., the cascade failure phenomenon, which ultimately leads to the interruption of the communication of the whole UAV swarm. The process is shown in Figure 3. First, a node in the communication layer network is attacked by a virus, and then the virus spreads through the communication between nodes in the network, and eventually all the nodes in the communication layer are invaded by the virus, which then leads to the failure of the whole communication layer network.

This section models the node failure propagation in the UAV swarm communication layer based on the SIR model. It is assumed that the UAV swarm system consists of *n* UAVs, each of which is in an initial susceptible state. When *t* = 0 is present, there are m infected state nodes in its communication layer. Therefore, the probability of an infected node infecting a susceptible node over time is *β*, and the probability of an infected node entering the recovery state is *γ*. The specific concepts of the model are as follows:
(1)Susceptible state node *S*: the node is not infected but lacks immunity and is susceptible to infection when exposed to an infected state node;(2)Infected state node *I*: the node has been infected by the virus, i.e., it fails and spreads the virus to neighboring nodes;(3)Recovery status node *Re*: The node has resumed its operational status. This status node will not be infected with a virus or have an impact on other nodes;(4)Infection rate *β*: the probability of susceptible node *S* being infected after contact with infected node *I*;(5)Recovery rate *γ*: the probability of an infected node *I* entering recovery state *Re*.


The information association process of the UAV swarm network is as follows:
Step 1:when *t* = 0 is set, randomize m infected nodes;Step 2:determine whether the population reaches the infection rate *β*, i.e., *m*/*n* > *β*. If it is not reached, randomly select a node from the neighboring nodes of the infected node to enter the infected state. Otherwise, go to Step 3;Step 3:Determine whether the recovery rate of the infected node reaches *γ*. If yes, the node is converted to a recovery state, and the algorithm ends. Otherwise, go to Step 4;Step 4:make *t* = *t* + 1 and repeat Steps 2 and 3 until there are no remaining nodes in the communication layer.


## 3. Methods for Assessing the Resilience of UAV Swarm Systems

The resilience assessment of UAV swarms provides a more unique perspective than other performance assessment methods. Compared to other properties such as robustness, resilience considers the UAV swarm state more comprehensively and is no longer limited to destruction resistance analysis of damage-absorbing capacity, which facilitates the construction of more resilient UAV swarm structures. Therefore, based on the above modeling process considering dynamic evolution, this section establishes a resilience assessment index for UAV swarm networks, proposes two different attack and recovery strategies, and proposes an algorithm for UAV swarm resilience assessment considering dynamic evolution.

### 3.1. The Concept of UAV Swarm Resilience

Due to the dynamic reconfiguration characteristics of UAV swarm systems, the introduction of the concept of resilience can better assess the state of UAV swarm systems. UAS resilience refers to the ability of the UAV swarm to absorb damage and minimize losses promptly after being disturbed or damaged, and to recover its performance to the expected optimal level as soon as possible to enhance the mission accomplishment.

The classical performance change curve of the UAV swarm during the disruption process is shown in Figure 4.

The curves represent the dynamic changes in system performance as two phases, destruction and recovery, based on the point in time when the destruction event occurs and the point in time when the recovery occurs. The horizontal coordinate represents the operation time *t*, and the vertical coordinate represents the UAS performance *y*(*t*).

The key time points and performance values in Figure 4 are defined below:

At $t_{0}$, the UAV swarm maintains normal peak performance ${\;y}_{D}$;

At $\;t_{d}$, the UAV swarm begins to be damaged, and the performance of the UAV swarm begins to degrade;

At ${\;t}_{\min}$, the performance of the UAV fleet decreases to the lowest level of stabilization at ${\;y}_{\min}$;

At ${\;t}_{r}$, the UAV swarm gradually recovers its performance by self-organizing and executing a recovery strategy;

At $\;t_{s}$, the UAV swarm returns to a steady state ${\;y}_{R}\;$ until $\;t_{final}$.

### 3.2. Extraction and Analysis of Resilience Indicators

Most studies of network systems rely on common statistical properties to determine basic measures of relevant topological features. Statistical properties of complex networks include node degree and degree distribution, clustering factor, and average path length. In this paper, the metrics are improved and adapted accordingly to the double layer network coupling relationship between the communication and structural layers of the UAV swarm.

Statement: the various mathematical notations and symbols defined are presented in Table 2 under Section 3.2 for the convenience of the readers.

#### 3.2.1. Node Degree and Degree Distribution

The degree is a key attribute of a network node that indicates the number of links between that node and other nodes. The degree distribution $p_{k}$ denotes the probability of a randomly selected node in the network with degree *k*. Since $p_{k}$ is a probability, it must satisfy the normalization constraint,
(1)$$\sum_{k=0}^{\infty }p_{k}=1.$$

Since the UAV swarm model established in this paper is a double layer expression model that considers the coupling of communication and structural layers, the node degree in the network model can be discussed in two categories:

(1) Calculate the degree value of the node $V_{qi}$ from the node links within the network layer; *N* denotes the number of active nodes in the UAV swarm network:(2)$$k_{qi}{(G}_{q})\;q=a,b,\;i=1,2,\ldots ,N;$$

(2) Calculate the degree value of node $V_{qi}$ from the node links between network layers:(3)$$k_{qi}^{l}{=1,V}_{qi}\in G_{a}{|G}_{b},$$
then the node degree of any node in the UAV swarm network can be expressed as:(4)$$k_{qi}{=k}_{qi}{(G}_{q}{)+k}_{qi}^{l},$$
the average degree in a UAV swarm network can be expressed as:(5)$$K=\frac{1}{2*N}\sum_{q=a}^{b}\sum_{i=1}^{N}k_{qi},$$
and the cumulative degree distribution in the UAV swarm network *P*(*k*) can be expressed as:(6)$$P(k)=\sum_{k\prime \geq k}p(k\prime ).$$

#### 3.2.2. Clustering Factor

The clustering factor indicates how densely the neighboring nodes of a node are linked to each other. For a node $V_{qi}$ of degree $k_{qi}$, $L_{qi}$ denotes the number of links between $k_{qi}$ neighbors of the node, and its local clustering factor is defined as:(7)$$C_{qi}=\frac{{2L}_{qi}}{k_{qi}{(k}_{qi}\;-\;1)}.$$

Similarly, local clustering factors in network models can be discussed in two categories:

(1) Calculate the local clustering factor of node $V_{qi}$ from the node links within the network layer:(8)$$L_{qi}{(G}_{q})\;q=a,b,\;i=1,2,\ldots ,N;$$

(2) Calculate the local clustering factor of the node $V_{qi}$ from the node links between the network layers:(9)$$L_{qi}^{l}{=1,\;V}_{qi}\in G_{a}{|G}_{b},$$
the local clustering factor of any node $V_{qi}$ in the UAV swarm network can be expressed as:(10)$$L_{qi}{=L}_{qi}{(G}_{q}{)+L}_{qi}^{l},$$
and the degree of clustering of the entire network of the UAV swarm network can be portrayed by the average clustering factor of all its nodes, *C* is defined as:(11)$$C=\frac{1}{2*N}\sum_{q=a}^{b}\sum_{i=1}^{N}C_{qi}.$$

#### 3.2.3. Average Path Length

The average path length is the average distance between all pairs of nodes in the network, which indicates how tightly linked the network is and can be defined by the arithmetic mean of the shortest paths between any pair of nodes in the network:(12)$$d=\frac{2}{N(N+1)}\sum_{\begin{array}{c} i,j=1,\ldots N\\ i\neq j \end{array}}d_{ij}.$$

#### 3.2.4. Network Efficiency

The efficiency of two nodes *i* and *j* in the network is denoted as the reciprocal of their distance, $1/d_{ij}$. Therefore, the efficiency of the network constructed in this paper is expressed as the average of the efficiency of each node in the communication layer.
(13)$$E=\frac{1}{N(N-1)}\sum_{i\neq j}\frac{1}{d_{ij}}.$$

#### 3.2.5. Comprehensive Evaluation Indicators

When the UAV fleet is facing different mission scenarios, the resilience metric needs to be appropriately adjusted. To characterize the dynamic process of resilience assessment, the weights of each indicator need to be set objectively when determining the comprehensive resilience assessment metric.

The method of independence weighting factors uses multiple regression analysis to calculate the factor of complex correlation and thus determine the weights. The larger the calculated compound correlation factor, the more repetitive information the indicator has, i.e., the smaller the weight. The factor of complex correlation reflects the amount of repeated information between indicators, while the weight reflects the amount of information contained in each indicator that is different from that of other indicators, so it can be used to evaluate indicators with complex relationships or to exclude the study of repeated information between evaluation indicators.

Assuming that there are *m* indicators, respectively,$\;x_{1}$,$\;x_{2}$, …,$\;x_{m}$, the greater the factor of complex correlation between the indicator $x_{j}$ and each of the other indicators, the stronger the linear relationship between $x_{j}$ and the other indicators. It indicates that $x_{j}$ can be represented by a linear combination of the other indicators; the greater the amount of repetitive information, the smaller the weight of the indicator.

The factor of complex correlation is calculated as:(14)$$R_{j}=\frac{\sum_{j=1}^{m}{(x}_{j}-\overset{-}{x})(\tilde{x}-\overset{-}{x})}{\sqrt{\sum_{j=1}^{m}{{(x}_{j}-\overset{-}{x})}^{2}{(\tilde{x}-\overset{-}{x})}^{2}}}\left(j=1,2,3,\ldots ,m\right),$$
where $\tilde{x}$ is the linear combination of the other variables in *x* excluding $x_{j}$ and $\overset{-}{x}$ is the average of *x*.

The inverse of the factor of complex correlation $R_{j}$ is selected and normalized to calculate the final weight value $w_{j}$:(15)$$R=\left[\;\frac{1}{R_{1}},\;\;\frac{1}{R_{2}},\;\frac{1}{R_{3}},\;\cdots ,\;\frac{1}{R_{m}}\;\right],$$
(16)$$W_{j}=\frac{\frac{1}{R_{j}}}{\sum_{j=1}^{m}\frac{1}{R_{j}}}.$$

Finally, a comprehensive resilience measure for the UAV swarm network can be obtained using the normalization method:(17)$$R=\sum_{\;j=1}^{m}w_{j}\frac{x_{j}-\min(x_{j})}{\max(x_{j})-\min(x_{j})},$$
where *R* represents the resilience of the UAV cluster network, R∈[0, 1].

### 3.3. Resilience Assessment Strategy Options

The resilience assessment of a UAV swarm is mainly divided into two phases: absorbing perturbation and recovering performance. The removal of nodes in the disturbance absorption phase and the dynamic reconfiguration in the performance recovery phase are the two important components of the resilience assessment, while the resilience assessment strategy is the method of removing nodes and recovering nodes. Therefore, the selection of different strategies may have a large impact on the assessment results of the UAS system.

Currently, there are two main types of attacks: random attacks and deliberate attacks. Random attacks can be manifested in two ways in the UAV swarm system: (1) simulating physical attacks, i.e., randomly removing nodes $V_{bi}$ in the structural layer, and at the same time, the link nodes $V_{ai}$ in the communication layer are also removed; and (2) simulating wireless network attacks, i.e., randomly removing nodes $V_{ai}$ in the communication layer. Deliberate attacks may be expressed as the removal of nodes based on UAV swarm network topology features, such as the removal of nodes based on the degree of the nodes in the network from largest to smallest.

Correspondingly, the recovery types are likewise divided into two types: random recovery, where the remaining nodes are randomly selected to link with the orphaned nodes, and purposeful recovery, where the remaining nodes are linked with the orphaned nodes based on their characteristic metrics, such as node degree size.

### 3.4. Resilience Assessment Algorithm

#### 3.4.1. Algorithm for Resilience Assessment Considering Dynamic Reconfiguration

Based on the dynamic reconfiguration network modeling in Section 2.2.1, the UAV swarm resilience assessment algorithm considering dynamic reconfiguration is pro-posed, and the pseudo-code of the algorithm is as Algorithm 1.
Input: double layer UAV swarm network and node set, including communication layer and structural layer;Output: dynamically reconfigured double layer UAV swarm network and a set of resilience indicators;Algorithm steps.


Communication layer node removal and dynamic reconfiguration:
(1)Randomly select a node from the communication layer and remove it from the communication layer;(2)Handle isolated nodes: if the removed node is isolated (not connected to any other node), select another node and establish a communication edge between them;(3)Repeat the above steps until all nodes are processed.


Structural layer node removal and dynamic reconfiguration:
(1)Randomly select a node from the structural layer and simultaneously remove it from both the communication and structural layers;(2)Handle isolated nodes: if the removed node is isolated, select another node and establish two edges between them in both the communication and structural layers;(3)Repeat the above steps until all nodes are processed.


**Algorithm 1:** Algorithm for resilience assessment considering dynamic reconfiguration**Input:** double layer UAV swarm network Ga, Gb; node set Sq(q=a,b)**Output:** dynamically reconfigured double layer UAV swarm network Ga, Gb; resilience indicators set1: node removal and dynamic reconfiguration in communication layer Ga: 2: **while** node number > 0 **do**3:  randomly or targetedly, choose nodes Vak from Sa4: delete Vak from Sa5: **for** each node Vai in Sa **do**6:  **if**
Vai an isolated node **then**7:   randomly or targetedly, choose node Vaj from Sa8:   add edge between Vai and Vaj9:  **end if**10: **end for**11: node removal and dynamic reconfiguration in structural layer Gb: 12: randomly or targetedly, choose nodes Vbk from Sb13: delete Vbk from Sb14: delete Vak from Sa15: **for** each node Vbi in Sb **do**16:  **if**
Vbi an isolated node **then**17:   randomly or targetedly, choose node Vaj from Sa18:   add an edge between Vbi and Vbj19:   add edge between Vai and Vaj20:  **end if**21: **end for**22: **end while**23: **return** (Ga, Gb) and *R*

#### 3.4.2. Algorithm for Resilience Assessment Considering Information Correlation

According to Section 2.2.2, the UAV swarm resilience assessment algorithm considering information correlation is proposed, and the pseudo-code of the algorithm is as Algorithm 2.

Input: double layer UAV swarm network, infection rate *β*, recovery rate *γ*:
Initialization: susceptible nodes *S*[*x*], infected nodes *I*[*z*], initial time *t* = 0;Output: double layer UAV swarm network with information correlation and a set of resilience indicators;Algorithm steps:
(1)Targeted select some nodes in the network, label them as infected nodes, and simultaneously remove them from susceptible nodes;(2)Infection and recovery process: loop through the network; for each infected node, randomly select other nodes for infection based on the infection rate β, and randomly decide whether the node recovers based on the recovery rate γ;(3)Resilience indicator calculation: calculate the network’s resilience indicators after each loop;(4)Return results: return the adjusted network and set of resilience indicators when there are no susceptible or infected nodes in the network.



**Algorithm 2:** Algorithm for resilience assessment considering information correlation**Input:** double layer UAV swarm network Ga, Gb; infection rate *β*; recovery rate *γ***Initialization:**
*S*[*x*], *I*[*z*], *t* = 0**Output:** double layer UAV swarm network Ga,Gb with information correlation and resilience indicators set 1: **for** *p* = 1 to *m* **do**2: targeted chose nodes Vai from Ga3:  add Vai to *I*[*z*]; *z* = *z* + 14:  delete Vai from *S*[*x*]5: **end for**6: **while** *S*[*x*] or *I*[*z*] not empty do7: **for** *i* = 1 to *z* **do**8:  generate a random number *r*9:  **if** *r* < *β*
**then**10:   targeted choose nodes Vaj from *S*[*x*]11:    add Vaj to *I*[*z*]; *z* = *z* + 112:   delete Vaj from *S*[*x*]13:  **end if**14: **end for**15:  **for** each node Vak in *I*[*z*] **do**16:  generate random number *r*17:  **if** *r* < *γ*
**then**18:   delete Vak from Ga, *I*[*z*]; *z* = *z* − 119:  **end if**20: **end for**21: calculate the comprehensive resilience metric *R*22: t=t+123: **end while**24: **return** (Ga, Gb) and *R*

### 3.5. Algorithm Flow

According to the self-organized double layer coupled network model of the UAV swarm established in this paper, three topologies of the sub-cluster network, random network, and BA network are constructed, and three resilience indicators of different network structures are statistically analyzed, and a comprehensive resilience assessment index is proposed. Two modes, random and directed, are adopted for the attack and recovery strategies of the network, which mainly contain two parts: (1) resilience assessment by attacking the double layer coupled network and considering the recovery process of dynamic reconfiguration, and (2) resilience assessment by attacking the communication layer of the network based on the improved node failure propagation model and considering the information correlation characteristics of the UAV swarm.

The flow of the resilience assessment method proposed in this paper, considering dynamic evolution, is shown in Figure 5.

## 4. Experimental Validation

In this section, based on the NetworkX library environment, we analyze the effects of different structures and different evaluation strategies on the resilience of UAV swarms based on the aforementioned establishment of a double layer coupled network model that considers the dynamic evolution of UAV swarms.

### 4.1. Modeling of a Double Layer Coupled Network

To better analyze the effects of different topologies on the resilience of UAV swarms, three complex network models are constructed based on the typical structure of UAV swarm self-organizing networks, and their visualization models and their respective adjacency matrices are shown in Figure 6, among which Figure 6a is a leader-follower-based sub-cluster structured network [30], which is set to be composed of 55 small UAVs. There are five leader UAVs in the system, and each leader has 10 follower UAVs to form a cluster structure, forming a total of five clusters, and the cluster heads (five leader UAVs) are connected through the communication layer. Figure 6b,c show the random network and BA network models with the same number of nodes and connected edges.

To analyze the differences further quantitatively between different network structures and explore the intrinsic connection between UAV swarm network structure and resilience assessment, the degree distribution is used to reflect the links between the entire network nodes of the UAV swarm, and the histograms of the degree distribution of the three network structures are shown in Figure 7.

From Figure 7, it can be seen that there are significant differences in the degree distributions among the three network structures with the same number of nodes and connecting edges, in which the node degrees of the cluster network model established in this paper are concentrated in four and five, the node degree distributions of the stochastic network model are less fluctuating, with most of them concentrated in 4–7, and the node degrees of the BA network model are more varied individually, with most of them concentrated in 3–5, and at the same time there are also individual cases with an extremely large degree of the nodes.

The specific characteristic data of the three network structures are shown in Table 3. Since the number of nodes and the number of connected edges are the same, the difference in the average degree of the three network structures is small. The average path length and average clustering coefficient of the cluster network model are the largest because the nodes are more densely linked to each other, and the average clustering coefficient of the random network model is the smallest because the nodes are more evenly distributed.

### 4.2. Resilience Assessment

#### 4.2.1. Resilience Assessment Methods Considering Dynamic Reconfiguration

In this section, based on the UAV swarm resilience assessment algorithm considering dynamic reconfiguration described in Section 3.4.1, two resilience assessment strategies, namely random attack and deliberate attack on nodes, are selected to assess and analyze the resilience of the above three different double layer coupled networks.

First, three topological indicators, namely, average node degree *k*, average clustering coefficient *C,* and network efficiency *E*, are selected to constitute the comprehensive assessment indexes of network resilience in this paper. Simulation experiments are conducted under the above-mentioned benchmark indexes, and all the experimental results are based on the average values of 30 independent runs. In this section, by analyzing 12 scenarios under different network topology types and different policy combinations, the changes of the three metrics under different scenarios are investigated, and the experiments simulate the network attack and recovery process in 300 steps with 0.1 s as the time step, and the results are shown in Figure 8.

Assuming that the UAV swarm is in a stable state before being attacked, the network is attacked within the 30th–140th steps and one node in the network is removed in each step. As the nodes in the network are removed one by one, the number of nodes decreases, leading to damage to the network structure, and therefore the values of the three metrics are gradually decreasing. To observe the effect of the recovery strategy, the network recovery process is set to start at the same time, 50 steps after the end of the attack, until the network stabilizes after the recovery is completed. After the recovery process starts, the nodes added to the network are instantly linked to the isolated nodes and island nodes, and the distribution of nodes in the network is relatively centralized at this moment, thus the values of the three metrics increase rapidly, and the performance of the network fluctuates and decreases slightly with the increase in the number of nodes, so the values of the three metrics may jump and then remain relatively stable.

Based on the independence weight factor method described above, the weights of the three resilience topology metrics $w_{1}{,\;w}_{2}{,\;\mathrm{and}\;w}_{3}$ ($w_{1}{+w}_{2}{+w}_{3}=1$) are determined, and the combined resilience metric *R* is obtained according to Equation (17). The performance changes of the three network structures under different strategies are shown in Figure 9. Under random attack, the trend of failure process of the three network structures is similar, and the performance decline trend is slightly slower than the rest of the networks due to the more average linking situation of the nodes of the random network; under deliberate attack, the resilience of the random network is better than the rest of the two networks, and the nodes with a high number of links in the network are more prone to fail under deliberate attack due to the preferred linking characteristics of the scale-free network, so the network declines the fastest. Under stochastic recovery, the random network has the fastest but the worst recovery, and the clustered network can recover better to its peak performance; under targeted recovery, the random network has the best recovery instead, and the BA network has the worst recovery.

#### 4.2.2. Resilience Assessment Methods Considering Information Correlation

The resilience of the above three double layer coupling networks is evaluated according to the UAV swarm resilience assessment algorithm, considering the information correlation described in 2.2.2.

Firstly, all the nodes of the UAV swarm network are set to be susceptible; in addition, the infection rate β=0.1 and the recovery rate γ=0.1 are set; similarly, the average node degree *k*, the average clustering coefficient *C*, and the network efficiency *E* are selected as three topological indicators as a comprehensive assessment of the resilience, i.e., the source nodes of the network infections under the SIR model are comprehensively selected based on the above indicators. Under the above benchmark indicators, simulation experiments are carried out because the virus will be directed to attack a node in the network, so the results of the experimental resilience changes in this section are based on the deliberate attack and target recovery strategy, and all the experimental results are based on the average of 30 independent runs, and the resilience curves of the three kinds of networks are obtained, as shown in Figure 10. As can be seen from the figure, the resilience curves of these three UAV swarm network structures do not differ much in general, and the resilience of the sub-cluster network is relatively better.

The effect of network topology on the resilience of UAV swarms was analyzed in the above process, and next this paper will analyze the effect of the γ parameter on the resilience of the network, as shown in Figure 11.

Through the analysis, it is found that the change trends of the three network structures are the same, so the setting of the γ parameter is within a reasonable range, which can effectively realize the attack and recovery processes of the network. As *γ* increases, the severity of network failure noticeably decreases, reaching a stable state for the first *γ* parameter. Table 4 displays the specific parameter settings along with the corresponding minimum and maximum values of network performance.

## 5. Conclusions

This paper addresses the increasing resilience demands for UAV swarm systems. It builds upon existing UAV swarm system modeling frameworks and constructs three typical UAV swarm double layer coupling network models based on complex networks. The study analyzes the dynamic reconfiguration and information correlation characteristics observed in UAV swarm application scenarios. It establishes a comprehensive resilience metric that considers multiple network topology indices. Additionally, the paper introduces an algorithm for assessing UAV swarm resilience, considering the dynamic evolution of the UAV swarm. This algorithm facilitates UAV swarm resilience assessment under different strategies. The main conclusions are as follows:
(1)The double layer coupling model of a UAV swarm complex network with three typical structures is proposed, which fully takes into account the different characteristics of the communication and structural layers of the UAV swarm self-organized network and can realize an effective simulation of the changes in the structure of the UAV swarm under the complex and changeable mission environment;(2)The proposed UAV swarm comprehensive resilience metric adopts three network topology indicators and determines their weights by the independence weight coefficient method, realizing the multidimensional extraction of the comprehensive metric and comprehensively and objectively assessing the system resilience;(3)Two different resilience assessment methods are proposed, in which the UAV swarm resilience assessment considering dynamic reconfiguration has a strong correlation with the network structure design and analyzes the resilience differences in the three network structures under different combinations of strategies; the UAV swarm resilience assessment considering the information correlation verifies the network resilience changes in the process of cascading failures induced by virus propagation and gives the resilience curves of the three network structures; and also the effects of different recovery rates on network resilience are analyzed. The resilience assessment results can be further used in the structural design and decision-making scenarios of UAV swarms.


## Figures and Tables

**Figure 1 sensors-24-00011-f001:**
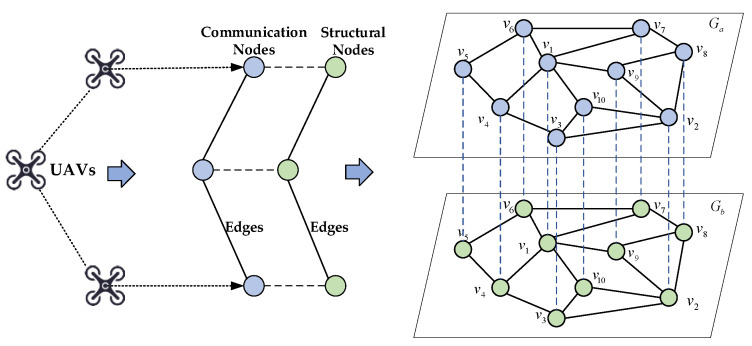
Construction of a double layer coupled UAV network.

**Figure 2 sensors-24-00011-f002:**
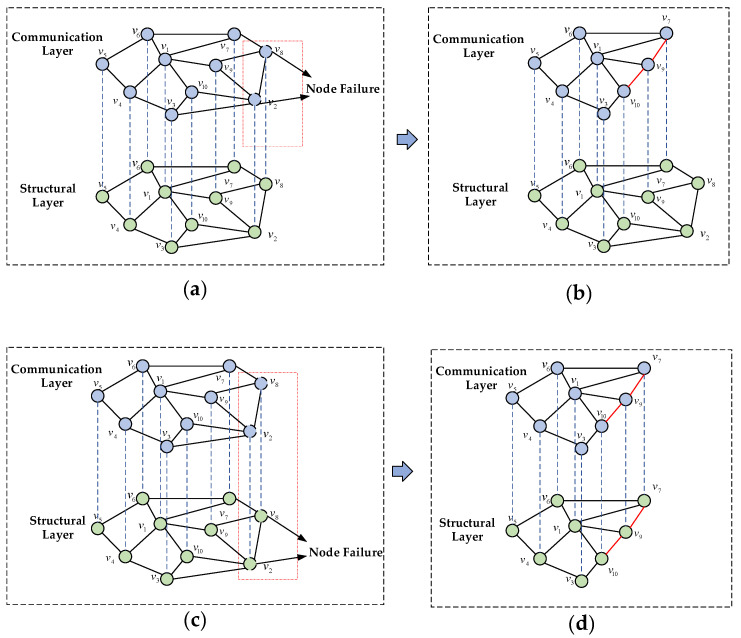
UAV node failure maintenance: (**a**) communication layer node failure; (**b**) communication layer dynamic reconfiguration; (**c**) structural layer node failure; (**d**) structural layer dynamic reconfiguration.

**Figure 3 sensors-24-00011-f003:**
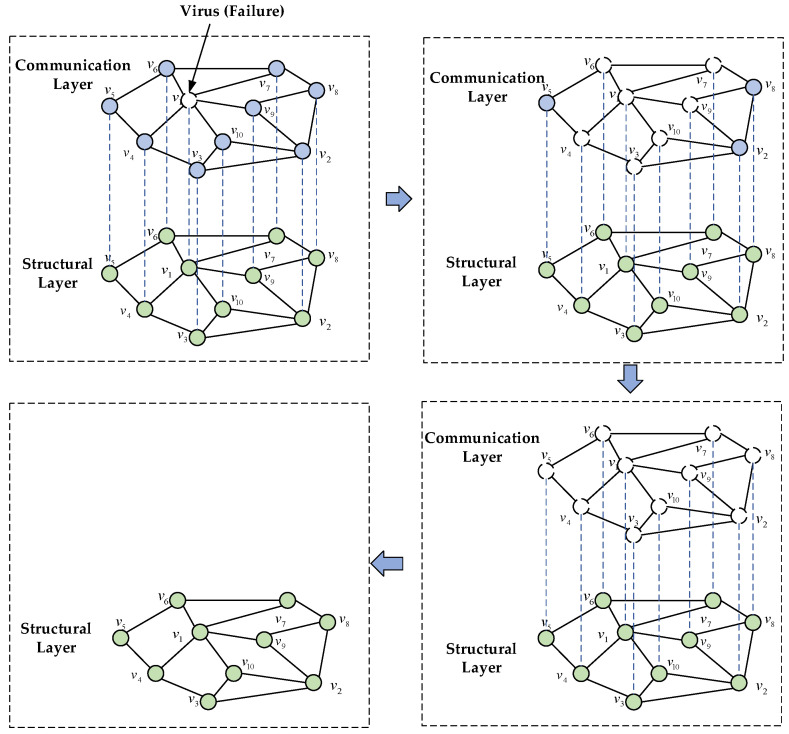
Cascade failure of UAV communications.

**Figure 4 sensors-24-00011-f004:**
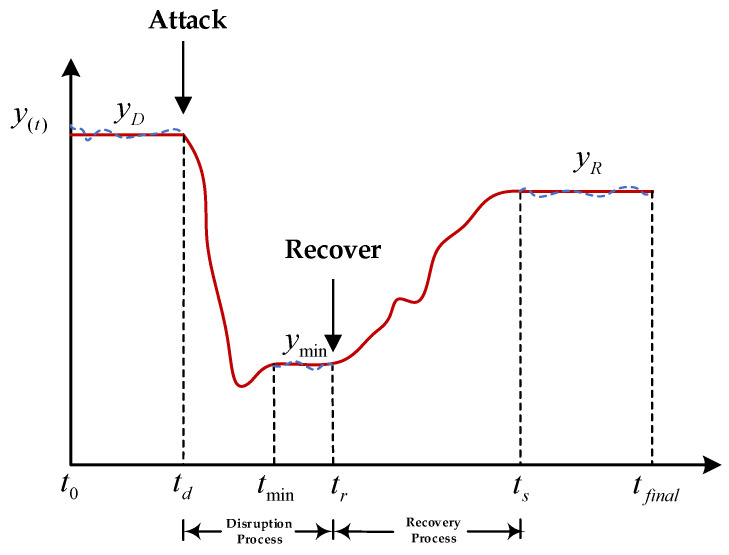
Performance change curve of UAVs.

**Figure 5 sensors-24-00011-f005:**
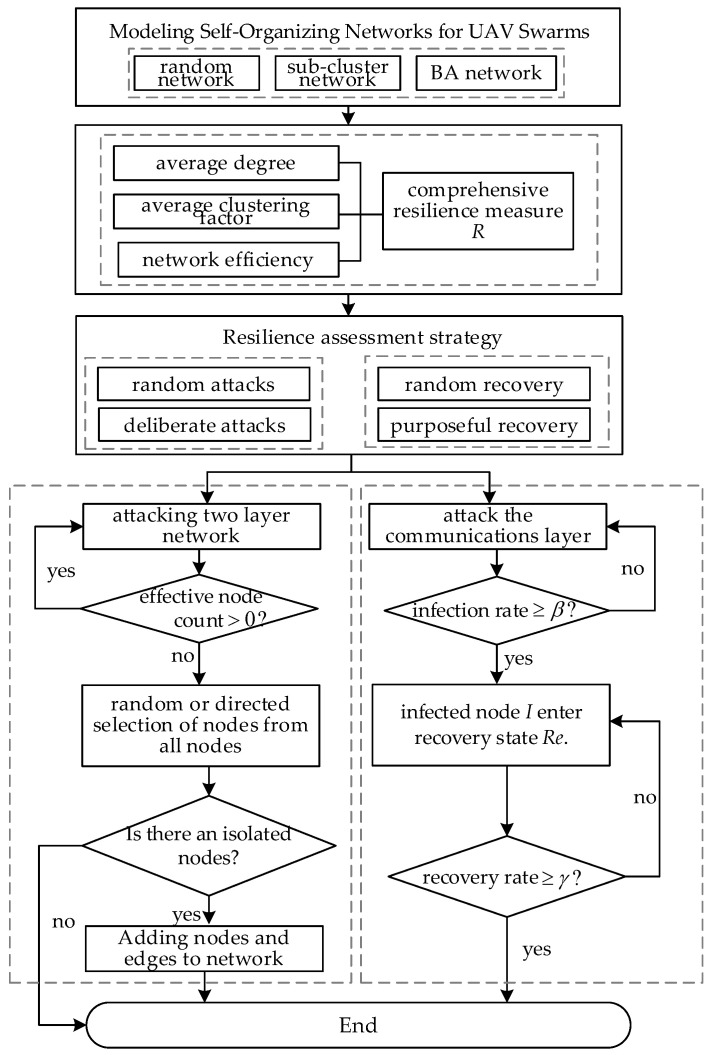
Algorithm flow of resilience assessment.

**Figure 6 sensors-24-00011-f006:**
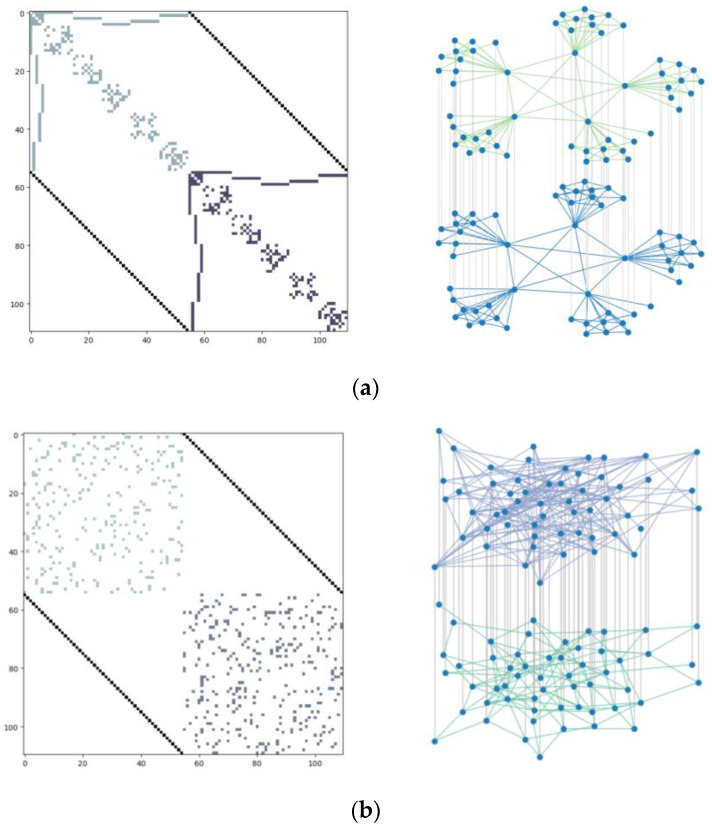
Adjacency matrix and coupling structure of the three double layer network models: (**a**) cluster network; (**b**) random network; (**c**) BA network.

**Figure 7 sensors-24-00011-f007:**
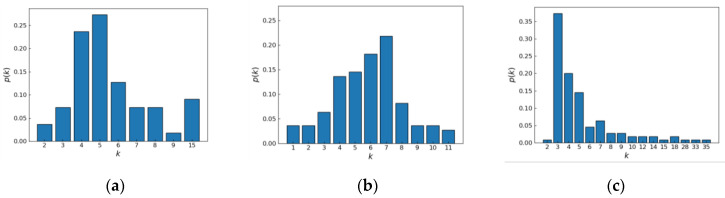
Degree distributions of three network structures: (**a**) cluster network; (**b**) random network; (**c**) BA network.

**Figure 8 sensors-24-00011-f008:**
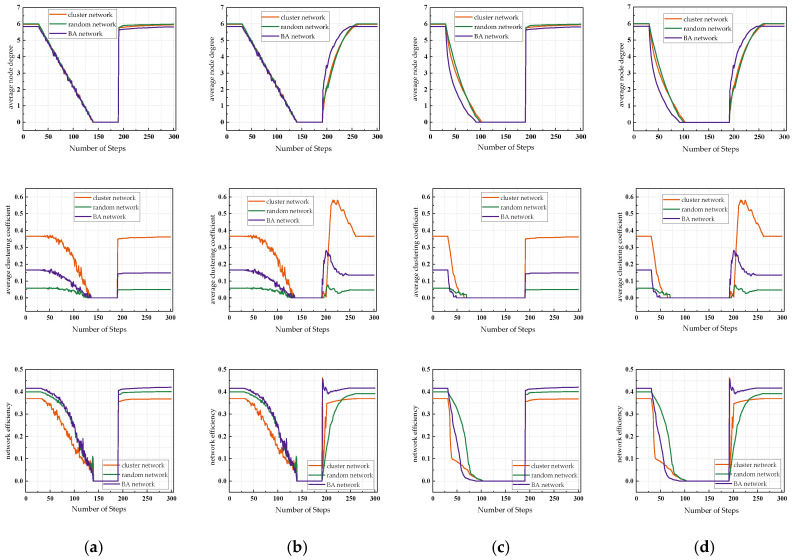
Variation in network performance of the UAV swarm under different strategies: (**a**) random attack + random recovery; (**b**) random attack + target recovery; (**c**) deliberate attack + random recovery; (**d**) deliberate attack + target recovery.

**Figure 9 sensors-24-00011-f009:**
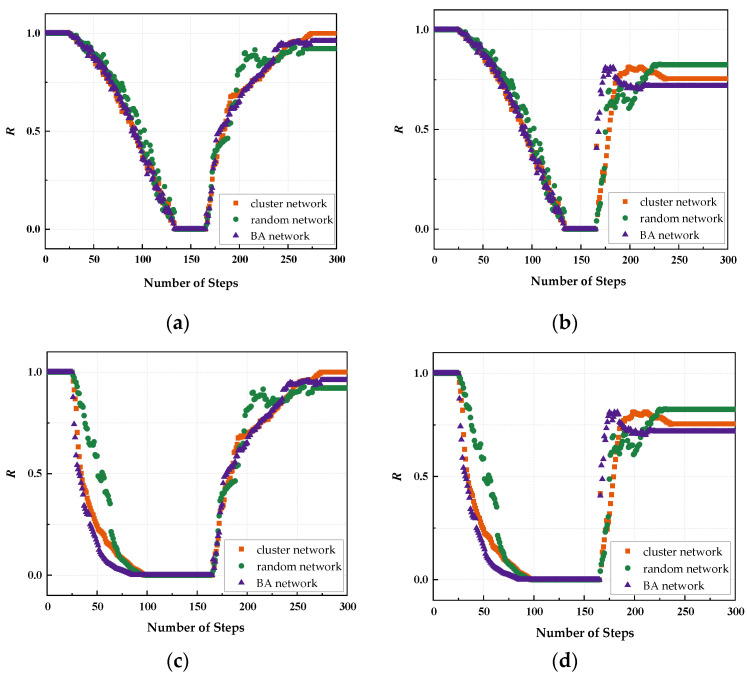
Three network resilience curves considering dynamic reconfiguration: (**a**) random attack + random recovery; (**b**) random attack + target recovery; (**c**) deliberate attack + random recovery; (**d**) deliberate attack + target recovery.

**Figure 10 sensors-24-00011-f010:**
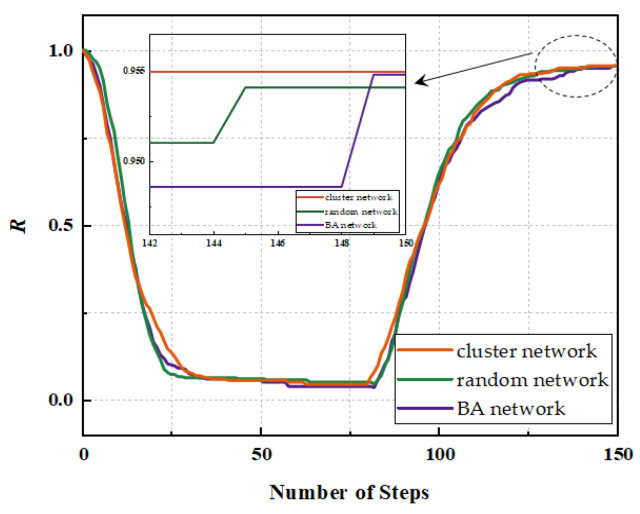
Three network resilience curves considering information correlation.

**Figure 11 sensors-24-00011-f011:**
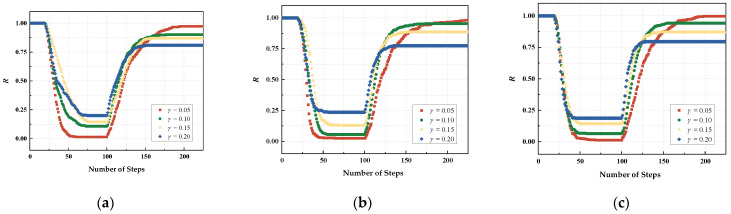
The impact of γ on network resilience: (**a**) cluster network; (**b**) random network; (**c**) BA network.

**Table 1 sensors-24-00011-t001:** Various methods for network modeling and resilience measurements.

References	Complex Network Models			Type of Destruction		Type of Recovery	
	Random Network	BA Network	WS (Watts–Strogatz) Network	Random Attack	Deliberate Attack	Random Recovery	Target Recovery
Tran [24]		✓		✓	✓	✓	✓
Cheng C [25]	✓	✓		✓		✓	✓
Bai G [26]		✓			✓		✓
Zhang P [27]	✓	✓	✓		✓	✓	

✓ means that the author have mentioned this method.

**Table 2 sensors-24-00011-t002:** Various mathematical notations and symbols are defined.

Various	Define
*k*	The degree value
$p_{k}$	The probability of a randomly selected node with degree *k*
*q*	The network layer; *q* = a indicates the communication layer; *q* = b indicates the structural layer
$V_{qi}$	The node *i* of network layer *G*(*q*)
$L_{qi}$	The number of links
$C_{qi}$	Clustering factor
*d*	Average path length
*E*	Network efficiency
*R*	Comprehensive evaluation indicator

**Table 3 sensors-24-00011-t003:** Indicators of the characteristics of different network structures.

Structure	Number of Nodes	Number of Edges	Average Degree	Average Path Length	Average Clustering Factor
(a)	110	329	5.982	3.062	0.366
(b)	110	329	5.818	2.836	0.047
(c)	110	329	5.997	2.614	0.120

**Table 4 sensors-24-00011-t004:** Comparison of network performance values with different parameters.

γ	**Network Performance Minimum**			**Network Performance Maximum**		
	**(a)**	**(b)**	**(c)**	**(a)**	**(b)**	**(c)**
0.05	0.0153	0.0244	0.0122	0.9728	0.9847	0.9969
0.10	0.1071	0.0552	0.0640	0.9013	0.9540	0.9421
0.15	0.1423	0.1256	0.1399	0.8669	0.8836	0.8692
0.20	0.2002	0.2347	0.1864	0.8089	0.7745	0.7951

## Data Availability

The data used in this paper can be requested from the corresponding authors upon request.

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
