# Peer review of "Research on UAV Swarm Network Modeling and Resilience Assessment Methods"

_sensors, 2023, doi:10.3390/s24010011_

Round 1

Reviewer 1 Report

Comments and Suggestions for Authors

The article provides a clear overview of the study's objectives, emphasizing the inadequacy of traditional UAV swarm assessment indicators and the need for a more comprehensive approach to resilience after system attacks. The construction of a double-layer coupled UAV swarm network model incorporating complex network theory is an intriguing contribution.

*The introduction section needs reorganization to effectively highlight the contributions of the study. It is recommended to divide the introduction into three subsections: Background of the Study, Related Works, and Contributions. Additionally, summarizing the related works in a table format and emphasizing the novel aspects of this study would enhance clarity and showcase the original contributions effectively.

*The use of three network topological indicators—average node degree, average clustering factor, and average network efficiency—demonstrates a comprehensive evaluation strategy. It would be beneficial to provide more details on how these indicators are calculated and the rationale behind choosing them over others for assessing resilience.

*Considering the dynamic nature of UAV swarm systems, it would be valuable to discuss potential extensions or future research directions. This might include addressing real-time adaptability, scalability concerns, or exploring the impact of various attack strategies.

*Review the article for precise language and grammar to enhance its readability and understanding.

Comments on the Quality of English Language

OK

Reviewer 2 Report

Comments and Suggestions for Authors

In this paper, a double-layer coupled UAV swarm network model is proposed, which considers the communication layer and the structure layer, and three network topology indexes representing the elasticity index of UAV swarm are proposed. On this basis, the resilience assessment method of UAV swarm considering dynamic evolution is designed.

This paper is good-writing, and the idea of this article is clear and enlightening. However, there are several issues of the poster to be further improved:

1.There are some typos and grammatical errors in this article, e.g.

second, most of the UAV swarm resilience indexes proposed at present are the improvement of some existing resilience assessment methods, and lack a Second, most of the proposed resilience metrics are improvements of some existing resilience assessment methods, lacking a more objective quantitative definition of resilience, and lack of a horizontal and all-round objective consideration;

and designs a resilience assessment method of UAV swarms that takes into account the dynamic evolution, so as to realize the resilience assessment of UAV swarms in multiple scenarios and under different strategies.

2.The subheadings of Chapter 2 are repeated and should be revised to highlight each section:

2.2.1. Dynamic Reconfiguration Analysis and Modeling

2.2.2. Dynamic Reconfiguration Analysis and Modeling

3.Adding the comparison results of existing algorithms to the simulation experiment can make the results more convincing.

Reviewer 3 Report

Comments and Suggestions for Authors

The authors in this paper have studied the modeling and resilience assessment methods for UAV swarm self-organized networks. To realize the resilience assessment of UAV swarm, the resilience assessment method for the UAV swarm considering their dynamic evolution is also designed in this paper. However, many issues including a lot of grammatical/typographical mistakes, use of long and compounded sentences throughout the paper are found, based on which the following suggestions are made to improve the contents of this paper. 

1.       Introduction should be rewritten to clearly state the important issues related to the UAV swarm self-organized networks and also the contributions of this study to address those issues.          

2.       Abbreviated terms like ‘BA’ given at line 67, ‘UAS’ used at line 221 must be expanded on its first use.

3.       Use the phrase ‘a single UAV’ instead of ‘single UAVs’ at line 30.

4.       Replace ‘link links’ with ‘links’ at line 64 and 84 also.

5.       The compounded sentence that begins with the phrase ‘Tran [25] model UAV swarms’ at line 63 and ends at line 70 must be divided into several simpler sentences.

6.       Replace ‘metrics.’ with ‘metrics,’ at line 70.

7.       Among various deficiencies of the current research on the UAV swarm, the second one stated at lines 84-88, must be rewritten to make it meaningful to the readers.

8.       The compounded sentence that begins with ‘In view of this,’ at line 96 and end at line 103 should be changed to multiple simple sentences.  

9.       Use the phrase ‘connections between the UAVs as edges’ instead of ‘connections between each UAV as edges’ at line 111.

10.   According some principle of modelling the UAV swarm stated at lines 119-120, i.e., “the physical distance between UAVs is regarded as an edge”, there will be an edge between each pair of UAVs in the UAV swarms in the structure layer. However, in the first part of figure 1, no edge is shown between the first and third UAVs in the structure layer although they are physically separated. So, the authors are suggested to either update the above said rule or the figure itself to make proper resemblance between them.   

11.   The compounded sentence stated at lines 135-141 should be rewritten or divided into simple sentences to make it meaningful to the readers.

12.   The phrase ‘and the node edges with dashed 155 lines indicate the damaged nodesgiven at lines 155-156 can be removed or rewritten as the failed/damaged nodes are shown in different way in Figure 2(a).

13.   In subsection 2.2.2, while describing the information association process of the UAV swarm network, it is said in step 2 “Determine whether the population reaches the infection rate β. If it is not reached, randomly select a node from the neighbor nodes of the infected node to enter the infected state.”  Here authors need to state what constitute the population and how the infection rate of the population is calculated. Another question is how a neighboring node can enter into the infected state if the population does not reach the infection rate.

14.   Various mathematical notations & symbols used in subsection 3.2, must be defined in a separate table at the beginning of the section. Authors should use single letter subscript instead of multi-letter subscript in the mathematical notations, e.g., use either ‘q’ or ‘i’ instead of ‘qi’ as subscript to represent the UAV node in the network.

15.   Algorithms 1 and 2 given in subsection 3.4 either should be elaborated or their different steps should be explained through illustration.

16.   ‘previous paper’ mentioned at line 345 must be cited there.

17.   ‘Average clustering coefficient’, which is taken to constitute the comprehensive assessment indexes of network resilience in this paper, should be defined.

18.   In all graphs given in figures 8, 9, 10 and 11, X-axis should be labeled with ‘Number of Steps’ as the experiments simulate the network attack and recovery process in 300 steps.

19.   The sentence given at lines 482-485, should be rewritten to make it meaningful.

20.   The very long and compounded sentence written at lines 490-497 must be divided into smaller and simple sentences to make it meaningful to the reader.

Round 2

Reviewer 3 Report

Comments and Suggestions for Authors

All my queries are well addressed except comment # 13 which includes multiple queries and so, each of them is required to be addressed in a separate paragraph.  In lines 209-210, the authors have stated that "If it is not reached, randomly select a node from the neighbor nodes of the infected node to enter the infected state". I am concerned about how a node from the neighboring nodes of the infected nodes can enter the infected state if the population does not reach the infection rate.

Author Response

I apologize for any confusion. Now, I'll provide a more detailed explanation:

In the SIR model, the unmanned aerial vehicle (UAV) swarm is divided into three states: susceptible (S), infectious (I), and recovered (R). The transmission of the infectious state occurs through direct contact, where nodes enter the infectious state by coming into contact with an infectious individual. In our model, "direct contact" can be understood as the presence of a link or edge between two nodes.

At the initial stage (t=0), m UAVs are randomly set to the infectious state (I), while the remaining UAVs are in the susceptible state (S). Therefore, in the next stage, if the infection rate is not reached, a node in the susceptible state (S) will transition to the infectious state (I) by randomly selecting one of its neighbors in the infectious state.

Should you have any further questions, please feel free to reach out to us.